# Can Leading by Example Alone Improve Cooperation?

**DOI:** 10.3390/bs14070601

**Published:** 2024-07-15

**Authors:** Ziying Zhang, Nguepi Tsafack Elvis, Jiawei Wang, Gonglin Hou

**Affiliations:** 1School of Psychology, Zhejiang Sci-Tech University, Hangzhou 310018, China; l20171030303@mails.zstu.edu.cn (Z.Z.); zzy1994058@sina.com (J.W.); 2School of Economics and Management, Zhejiang Sci-Tech University, Hangzhou 310018, China; l20171030203@mails.zstu.cn

**Keywords:** leading by example, public cooperation, incentive effect, public goods game

## Abstract

Cooperation is essential for the survival of human society. Understanding the nature of cooperation and its underlying mechanisms is crucial for studying human behavior. This paper investigates the impact of leadership on public cooperation by employing repeated sequential public goods games, as well as by examining whether leading by example (through rewards and punishments) can promote cooperation and organizational success. The leaders were assigned randomly and were given the authority to reward or punish. As a result, (1) the leaders showed a strong tendency toward reciprocity by punishing free riders and rewarding cooperators at their own expense, which enhanced the intrinsic motivation for others to follow their example; and (2) both rewards and punishments were effective in promoting cooperation, but punishment was more effective in sustaining a high level of collaboration. Additionally, leaders preferred using rewards and were more reluctant to use punishments. These findings are crucial for creating organizational structures that foster cooperation.

## 1. Introduction

When considering the selection of leaders, Drucker and Maciariello [1] emphasized the importance of integrity. They stated, “I would look for integrity. A leader sets an example, especially a strong leader. He or she is somebody on whom people, especially younger people, in the organization, model themselves”. Leadership behavior is crucial for promoting cooperation among people, which is essential for organizational success [2,3,4,5]. Therefore, the leader’s primary responsibility is motivating team members to work together in pursuit of the organization’s objectives [6,7]. However, it is difficult for each member to demonstrate willingness and behavior to cooperate as the inherent selfish motives of humans tend to lead people to be uncooperative, especially those who enjoy the fruits of collective efforts without contributing themselves [8,9]. Such free-riding behavior will affect people’s cooperation and even cause the organization to collapse. Bass [10] found that inspiring other members to transcend selfishness positively affected the level of cooperation and organizational efficiency. Hermalin [11] introduced the concept of leading by example and examined its motivational impact. He described leaders as individuals with followers who voluntarily choose to follow them, rather than being forced. Some researchers have also defined leading by example operationally. For instance, in the social dilemma game, the first participant who chose to contribute was considered the leader, while the others who followed suit were considered the followers [12,13,14,15].

Previous studies have found that, regardless of whether under a symmetric information situation (where all have information about the marginal returns of contribution) or under an asymmetric information situation (where only the leader possesses information about the marginal returns of contribution), the presence of the leader promotes the level of cooperation to some extent [16,17]. For example, Moxnes and Heijden [12] found that, when the leader took the lead and their choice was known to others, a significant incentive effect occurred in the public goods game. Gächter and Renner [18] investigated that the contributions of leaders and followers were highly correlated even in a one-shot game. Notably, the leader’s contribution was systematically higher than the followers’ [13,19,20]. This was consistent with the finding that, even if the cooperation could not maximize one’s own payoff and there was no long-term reputational interest, people were still willing to pay the price to promote cooperation [4,5,7,21,22]. Interestingly, when selected as leaders, the contribution of the former free riders would increase by 114%, indicating that the role of the leader promoted cooperative behavior [13].

When leaders lead by example, it can encourage cooperation. However, many studies have found that, as their experiments progressed, followers tend to encroach on the leader’s interests and become free riders. Consequently, the leader reduces their contribution, which gradually decreases the level of cooperation over time [12,13,15]. These findings show that, if the leader’s short-term and long-term interests are not guaranteed, the incentive effect on public cooperation cannot be sustained for a long time; thus, cooperation will eventually collapse. Therefore, the incentive impact of leading by example on public cooperation should be strengthened through an external incentive mechanism. Potters et al. [14] pioneered a study on empowering the leader. After giving the leader the freedom to assign the team’s output, the level of cooperation among the team members was effectively improved [17]. Güth et al. [15] examined the effect of punishment on group cooperation by giving leaders the right to expel any members during a specific period. Their study found that the right of expulsion could improve the team members’ cooperation level. In a study by Rivas and Sutter (2009), the research found that empowering the leaders with the right to expel had a greater impact on promoting cooperation compared to rewarding other members. Specifically, punishment had a better incentive effect. These studies reveal that giving the leader the power to control the members could effectively stop the members from taking advantage of the situation and protect the leader’s interests. This allowed the leaders to take the lead in making contributions without any worries. However, it was discovered that when free riders were expelled, all members suffered the consequences [15,23]. While the expulsion led to losses for the individuals who were kicked out, it also reduced the number of members contributing to the public resources in the next round, resulting in financial losses for all members.

In a study by Rivas and Sutter [23], a problem was found with the setting of the right to reward. The leader had the right to reward a sum of game tokens to one of the members after knowing everyone’s contribution. However, the game tokens used as a reward were contributed by the leader and other unrewarded members, meaning that the cost of the reward was borne by the other members of the group. As a result, the reward for one person became a punishment for other members, causing the cost of implementing the reward to exceed that of expulsion. This led to a decreased use of rewards by the leader. These findings were related to the experimental design, as the asymmetric reward and punishment settings did not put the two in the same position for comparison.

The objectives of this study were designed to shed light on the following questions: First, how does the presence of a leader impact the level of cooperation and payoff in a public goods game? Second, what are the effects of empowering leaders with the ability to reward or punish according to the level of public cooperation? Third, how do leaders’ contributions and payoffs compare to those of followers in different experimental conditions? Last but not least, what behavioral preferences do humans exhibit for punishment and reward in the context of promoting cooperation? These are the key questions that we endeavor to answer in this paper.

In order to achieve the research goals, this study aims to examine how leadership, through leading by example and the use of rewards and punishments, affects public cooperation in a repeated sequential public goods game. This study also investigates how leading by example, combined with the authority to reward or punish, impacts the level of cooperation and payoff within an organization. Furthermore, in this research, the aim is to also compare the impact of incentivizing leaders through rewards and punishments on cooperation levels within the same position.

To fix the problems with the study ideas, this research uses a strong reciprocal mechanism that is commonly employed in simultaneous public goods games as the method for both reward and punishment [7,24,25,26,27].

Based on the points mentioned above, we hypothesized the following results for the repeated sequential public goods game: (1) The presence of a leader would enhance cooperation levels. (2) The leader would contribute more and benefit less than the followers. (3) Empowering the leader with the ability to reward or punish would further promote and maintain cooperation. (4) The right to reward and the right to punish would have different effects on promoting cooperation levels.

## 2. Methods

This study’s methods section describes the experimental setup with 160 participants assigned to four groups: control, random leader, random leader with reward, and random leader with punishment. The experiment involved 32 rounds of a public goods game, with leaders in the latter two groups having the power to reward or punish.

### 2.1. Participants

The experiment involved 160 Chinese undergraduate and graduate students, comprising 77 males and 83 females. It was conducted between 2022 and 2023. None of them had taken part in any public goods experiment before. The participants were randomly assigned to four groups: the control (C) group, the random leader (RL) group, the random leader reward (RLR) group, and the random leader punishment (RLP) group. The 40 participants were randomly assigned to 10 teams in each group. In each team, all 4 participants completed the entire experiment together. The average payoff for the C group was CNY 11, and then CNY 12 for the RL group, CNY 13.5 for the RLR group, and CNY 13.5 for the RLP group. Before the experiments, all participants signed informed consent documents.

### 2.2. Procedures

After arriving at the laboratory, each participant extracted a small card tagging their room number, and they entered the separate and isolated rooms accordingly. The subjects were anonymous to each other, and the number (No. 1 to 4) displayed on the computer was different from the room number. Before the experiment, each participant received a printout of the instructions. The experimenter read it loudly and answered the participants’ questions. Participants also had to complete an eight-question test to understand the experiment’s rules. Appendix A, Appendix B and Appendix C provide the instructions and experimental interfaces for the control (C) group, the random leader (RL) group, the random leader punishment (RLP), or random leader reward (RLR) groups, respectively.

The experimental design was carried out under a repeated sequential public goods game. To fully investigate the role of the internal and external incentive mechanisms in improving the level of cooperation, we increased the experiment rounds to 32. A partner design was adopted, meaning that the same team of 4 participants completed the whole 32 rounds together.

As mentioned earlier, the present study included four groups. For the C group, participants played the standard public goods game for 32 rounds in teams, where the voluntary contribution mechanism was adopted. At the beginning of each round, each participant received 20 tokens. They were free to decide the number of tokens ($g_{i}$) that would be donated to the public account in any integer from 0 to 20 and then keep the rest. The donated tokens in the public account would be multiplied by 1.6 and then evenly distributed to each participant. In each round, the team participants contributed their tokens together and received feedback on the sum of tokens donated to the public account and their own payoff.

The payoff structure in a public goods game is defined as follows: The payoff ($\pi_{i}^{1}$) of each participant *i* given their contribution $g_{i}$ (and contributions of others) is calculated by the following:(1)$$\pi_{i}^{1}=20-g_{i}+0.4\;*\sum_{j=1}^{n}g_{j},\;i\;=\;1,\;\ldots ,n.$$

The payoff was not taken into account in the following round.

For the RL group, each round consisted of two stages. In the first stage, one participant was randomly selected as the leader by the experimental procedure and required to contribute their tokens ($g_{i}$) first. In the second stage, the other three participants were informed of the leader’s contribution and required to make their contribution decisions ($g_{i}$) at the same time. At the end of the second stage, the participants received feedback on the sum of tokens donated and their own payoff ($\pi_{i}^{2}$).
(2)$$\pi_{i}^{2}=20-g_{i}+0.4\;*\sum_{j=1}^{n}g_{j},\;i\;=\;1,\;\ldots ,n.$$

For the RLR group (i.e., the leaders have the right to reward others) and the RLP group (i.e., the leaders have the right to punish others), one more stage was included. In the same two stages as the RL group, the leaders were able to see the contributions of the other three participants in the second stage. They could then choose to keep their tokens or use them to reward $\left(R\right)\;$or punish $\left(P\right)$ the other three participants. The cost–benefit ratio for reward or punishment was 1:3, meaning that one token used by the leaders could increase or decrease the tokens of the other participants by three. At the end of the third stage, the participants received feedback on their actions and their payoff. Following the convention of Experimental Economics, the terms “increase” and “decrease” were used instead of “reward” and “punishment” in the experiment. The payoff of the leader and the followers can be calculated using the following formulas:(3)$$\mathrm{Leader}:\;\pi_{i}^{L}=\pi_{i}^{1}-P/R,\;i\;=\;1,\;\ldots ,n,$$

(4)$$\mathrm{Followers}:\;\pi_{i}^{F}=\pi_{i}^{1}\pm 3\times \left\{\begin{array}{c} P\\ R \end{array}\right.,\;i\;=\;1,\;\ldots ,n,$$
where *L* represents the leaders and *F* stands for followers.

It took an average of 30 min to complete the whole experiment. The total payoff of each participant in the experiment was the sum of the tokens earned in 32 rounds. These tokens were exchanged for CNY at the end of the experiment, with one token exchanged for CNY 0.015. After the experiment, the participants were asked to fill in a questionnaire, including demographic information, the reasons for making decisions in the experiment, etc. At the end, each participant would receive a monetary reward.

## 3. Results

We used z-Tree to edit and run the experimental procedure [28]. The research analyzed the contribution and the payoff using Statistical Package for the Social Sciences (SPSS) 25.0. Based on previous studies, we used the rank-sum test for between-group differences and the signed-rank test for within-group differences.

### 3.1. Comparison of the Random Leader Group and the Control Group

#### 3.1.1. Overall Contribution and Payoff

In the graph shown in Figure 1 below, the overall level of cooperation decreased over time for both the C and RL groups. Throughout the experiment, the RL group consistently exhibited higher and increasing contributions compared to the C group.

The results of the rank-sum test revealed that the average contribution of the RL group was significantly higher than that of the C group (*z* = −2.192, *p* = 0.028), as displayed in Table 1. Additionally, the average payoff of the RL group was also significantly higher than that of the C group (*z* = −2.192, *p* = 0.028), as shown in Table 1.

#### 3.1.2. Contribution and Payoff of the Leaders and the Followers

The results of the signed-rank test showed that the average contribution of the leaders was significantly higher than that of the followers in the RL group (*z* = −2.803, *p* = 0.005), as displayed in Table 1 above. A strong positive correlation was found between the average contributions of the leaders and the followers, with Spearman’s r = 0.903 and *p* < 0.001. Additionally, the average payoff of the followers in the RL group was significantly higher than that of the leaders, with *z* = −2.803 and *p* = 0.005 (refer to Table 1).

The random sum test results showed that the average contribution in the RL group was significantly higher than that of the C group (*z* = −3.477, *p* = 0.001, see Table 1). However, there was no significant difference between the average contribution of the followers and that of the C group (*z* = −1.512, *p* = 0.131). Additionally, the average payoff of the leaders in the RL group was significantly lower than that of the C group (*z* = −3.024, *p* = 0.002), while the average payoff of the followers in the RL group was significantly higher than that of the C group (*z* = −3.024, *p* = 0.002, see Table 1).

### 3.2. Comparison of the Two Groups with Leadership Power and the Random Leader Groups

#### 3.2.1. Overall Contribution

Figure 2 illustrates that the overall cooperation level of the two groups with leadership power (i.e., the RLR and RLP groups) was higher than that of the without-power group (the RL group) in 32 rounds. Additionally, the average contribution of the RLP group was higher than that of the RLR group in most rounds.

The results of the rank-sum test revealed that the average contribution of both the RLP group (*z* = −2.948, *p* = 0.003) and the RLR group (*z* = −2.721, *p* = 0.007) was significantly higher than that of the RL group. However, the difference in average contribution between the RLP group and the RLR group was not significant (*z* = 0.605, *p* = 0.545, see Table 2 below).

#### 3.2.2. Contribution of the Leaders and the Followers

The results of the signed-rank test showed that the average contribution of the leaders was significantly higher than that of the followers in both the RLR group (*z* = −2.701, *p* = 0.007) and the RLP group (*z* = −2.803, *p* = 0.005), as displayed in Table 2. Spearman rank correlation analysis revealed that the average contribution of the leaders and the followers were significantly positively correlated in both groups (RLR group: *r* = 0.857 and *p* = 0.002; RLP group: *r* = 0.915 and *p* < 0.001).

In addition, the results of the rank-sum test showed that the contributions of the followers in the groups with leadership power were significantly higher than that in the RL group without power (RLR vs. RL: *z* = −2.984 and *p* = 0.003; RLP vs. RL: *z* = −3.250 and *p* = 0.001, see Table 2).

#### 3.2.3. Overall Payoff

Although the use of rewards or punishment makes the average payoff in the third stage of the two with-power groups lower than that in the second stage (see Figure 3), the results of the rank-sum test revealed that the average final payoff of the two with-power groups in the third stage was still significantly higher than that in the C group (RLR vs. C: *z* = −1.814 and *p* = 0.07; RLP vs. C: *z* = −3.628 and *p* < 0.001). However, no significant difference was found when comparing the average final payoff of the two with-power groups with that of the RL group (RLR vs. RL: *z* = −1.814 and *p* = 0.07; RLP vs. RL: *z* = −1.361 and *p* = 0.174).

#### 3.2.4. Payoff of the Leaders and the Followers

Regarding the second stage, the results analysis of the signed rank-sum test showed that the average payoff of the leaders was significantly lower than that of the followers in both with-power groups (RLR: *z* = −2.701 and *p* = 0.007; RLP: *z* = −2.803 and *p* = 0.005, see Figure 4). However, the results of the rank-sum tests showed that the average payoff of the leaders in the two with-power groups was significantly higher than that of the RL group (RLR vs. RL: *z* = −3.477 and *p* = 0.001; RLP vs. RL: *z* = −3.704 and *p* < 0.001), and higher than that of the C group (RLR vs. C: *z* = −1.965 and *p* = 0.049; RLP vs. C: *z* = −3.250 and *p* = 0.001).

These findings indicate that the empowerment of leaders increased the average payoff of the leaders. Furthermore, the research analysis found that the average payoff of the followers in the two with-power groups was significantly higher than that of the RL group (RLR vs. RL: *z* = −2.495 and *p* = 0.013; RLP vs. RL: *z* = −2.343 and *p* = 0.019).

For the third stage, the use of punishment reduced the average payoff of the followers and the leaders, as shown in Figure 4. However, the average payoff of the leaders in the RLP group was still significantly higher than that of the RL group, but not different from that of the C group (rank-sum test, RLP vs. RL: *z* = −3.175 and *p* = 0.001; RLP vs. C: *z* = −1.587 and *p* = 0.112, see Figure 4). In addition, the use of rewards significantly reduced the average payoff of the leaders (rank-sum test, *z* = −3.099 and *p* = 0.002) and increased the average payoff of the followers (rank-sum test, *z* = −2.343 and *p* = 0.019), resulting in a significant difference between the leaders and the followers (sign rank-sum test, *z* = −2.803 and *p* = 0.005).

### 3.3. Reward and Punishment

As shown in Figure 5, the number of rewards was always larger than that of punishments in the 32 rounds, but the number of tokens used for reward and punishment was similar. The number of rewards steeply decreased in the first 10 rounds and then slowly until reaching a stable level. The number of punishments and the number of tokens used for punishment tended to be stable on the whole, with short-term upward trends in the early and middle rounds, which then declined to be stable.

The rank-sum test results showed that the number of reward usage was significantly higher than that of punishment usage (*z* = −2.612 and *p* = 0.009 (see Table 3)). Regarding the number of tokens used for reward or punishment, the rank-sum test results revealed no significant difference between the RLR group and the RLP group (*z* = −1.587 and *p* = 0.112 (see Table 3)).

In Figure 6 below, the number of single tokens used for rewards and punishments (which was obtained by dividing the average number of tokens used by the average number of power usage) in each round. The results of the Mann–Whitney U test showed that the number of single tokens used for punishments (*M* = 2.14 and *SD* = 0.66) was significantly higher than that for rewards (*M* = 0.4 and *SD* = 0.08): *z* = −6.879 and *p* < 0.001.

The results of the Spearman rank correlation analyses showed that the difference in the average contribution between the leaders and the followers was negatively correlated with the number of single tokens used for rewards (*r* = −0.721 and *p* = 0.019) but not with the number of rewards (*r* = −0.596 and *p* = 0.069). However, the difference in average contribution between the leaders and the followers was positively correlated with the number of punishments (*r* = 0.835 and *p* = 0.003) but not with the number of single tokens used for punishments (*r* = 0.055 and *p* = 0.881). These findings demonstrated that the difference between the average contribution of the leaders and the followers influenced the behavior of the leaders in issuing rewards and punishments. The greater the difference in contribution, the less the reward and the greater the punishment. Notably, leaders were likely to use fewer tokens for rewards, or they would increase the number of punishments as the difference in contribution increased.

In Figure 7 below, the less a follower contributed than a leader, the more likely they were to be punished. Only 8% of the followers with no contribution deviation were punished. Interestingly, as the followers’ contributions surpassed the average contribution, the probability of being punished also increased, indicating malicious punishments or revenge on some free-riding leaders.

Figure 7 also revealed that the more a follower contributed than a leader, the more likely they were to be rewarded. Notably, when the followers and the leaders contributed the same amount, the possibility of the followers being rewarded exceeded 70%, which was an even higher possibility than for those who contributed more than the leaders. The reason for this phenomenon was that many rewards were given when both the leaders and followers contributed all or most of the tokens, thus leading to a contribution deviation of 0.

## 4. Discussion

The present study compared leading by example with leadership using management power (i.e., by combining leading by example with a centralized external incentive mechanism), and the influence of a leader on organizational cooperation and benefits was also investigated. The results shed light on the mechanism that optimizes the incentive effect of leading by example.

### 4.1. The Influence of Leading by Example on the Level of Cooperation and Payoff

Our results demonstrated that the presence of leaders significantly improved the level of cooperation and payoff. A study by Gächter and Renner [18] found that being chosen as leaders visibly increased the cooperation level of free riders, while no such changes occurred for those who were already cooperative. Importantly, there was no difference in the level of cooperation between the two types of leaders and leaders who were randomly selected. This suggests that as long as there is a leader, public cooperation improves [13].

This research study discovered that the average contribution of the RL group varied greatly throughout the experiment. This could be related to the experiment’s design, in which the leader was randomly selected in each round, as this approach can be seen as a variation of designating leaders under a stranger design in the repeated sequential public goods game, which is an approach that exists between fixed leaders and rotating leaders [15].

Previous studies in the literature have indicated that individuals exhibit several stable behavioral patterns in these type of experiments: cooperators, who contribute at the Pareto optimal level; traitors or free riders, who contribute little; reciprocators or imitators, who contribute a part or multiples of the average of observed contribution; and strategists, who strategically chose their behavior to maximize their payoff [29,30,31,32]. The study by Fischbacher and Gächter [29] distinguished the cooperative preferences, and they then divided individuals into free riders, unconditional cooperators, and conditional cooperators. They found that most participants were conditional cooperators, who integrated their cooperative preferences and the cooperation level of others to determine their contribution. Dong et al. [33] also found that cooperative preferences significantly influenced the level of cooperation. They demonstrated that cooperative preference was a relatively stable behavioral pattern that is not easily influenced by external factors.

In this research study, the randomly selected leaders could have any of the cooperative preferences mentioned above. If the chosen leaders were highly inclined to promote cooperation, the average contribution was expected to increase. However, if they were not, the average contribution would decrease, causing fluctuations. Most individuals were conditional cooperators; therefore, when they acted as leaders, they preferred to select strategies that maximized their personal interests. Consequently, their willingness to lead declined over time, resulting in the progressive loss of the motivating effect of leading by example.

Our findings further confirmed that, while the presence of leaders improved the level of cooperation and the average payoff, it did not bring benefits to the leaders themselves. Leaders sacrificed their personal interests by contributing more and receiving less in order to encourage more people to behave cooperatively and reciprocally. However, the participants who did not contribute continued to benefit at the expense of the leaders and others, which led the leaders and cooperators to protect their interests by reducing their contributions continuously. As a result, the level of public cooperation and payoff gradually decreased over time. This highlights that leading by example not only enhances overall cooperation, but also improves the overall payoff. However, this incentive effect is limited and diminishes over time, indicating a marked diminishing marginal utility. While being a leader can promote cooperation, it may not bring direct benefits to the leader. On the contrary, followers benefit the most while the leader sacrifices personal interests to enhance the organization’s overall benefits. Over time, the leader may lose the willingness to lead the group. Therefore, it is necessary to protect the interests of leaders by improving the treatment of leaders in practice.

### 4.2. The Influence of External Incentive Mechanisms on the Incentive Effect of Leading by Example

Our results discovered that, compared with the RL group, empowering the leaders with the right of reward or punishment significantly increased the level of overall cooperation. Furthermore, the power of punishment was better than that of reward in maintaining a stable level of cooperation. Although there was still a significant difference in the level of cooperation between the leader and the followers in the two groups with power, the level of cooperation of the followers in the two groups was significantly improved compared with that of the followers in the RL group, indicating that empowering leaders further promoted the level of public cooperation.

The present study took a different approach compared to previous studies in using a strong reciprocity reward or punishment mechanism. Strong reciprocity refers to the tendency to incur personal costs to punish those who violate social norms (altruistic punishments) and to reward cooperators (altruistic rewards), even when the personal costs will not be repaid [34]. The results showed that the strong reciprocity reward or punishment mechanism, especially altruistic punishments, effectively maintained stable cooperation. Previous studies have also demonstrated that while both the power of reward and punishment can improve the level of cooperation, they play different roles. Punishment serves as a warning against social norms, whereas reward only hints at above-average cooperation. The same amount of reward cannot achieve the same impact as the warning effect of punishment [25,35]. In the long run, punishment can achieve a higher level of cooperation at a lower cost [36,37].

The promotion of the strong reciprocity reward or punishment mechanism, in addition to the incentive effect of leading by example, was reflected not only in the level of cooperation, but also in the payoff. Although there was no significant difference in the overall payoff between the RLR or RLP group and the RL group, further analyses in the second stage revealed that the payoff of the leaders in the two groups with power was significantly higher than that of the leaders in the without-power RL group. The same result patterns were evident in the payoff of the followers. Notably, in the second stage of each round, no reward or punishment was implemented. This means that the leaders’ ability to reward or punish team members themselves improved the payoff for both the leaders and the followers.

In the third stage, after the leaders implemented rewards or punishments, only the payoff of the followers in the RLR group increased. However, the payoff of the leaders in the two with-power groups and the payoff of the followers in the RLP group decreased. As a result, the difference in the payoff between the leaders and followers in the RLP group disappeared, and the payoff of the leaders in this group was significantly larger than that of the leaders in the RL group. This means that the power of punishment was more effective in safeguarding the interests of the leaders compared to the power of reward.

### 4.3. The Way the Leaders Used the Rewards and the Punishments

Our results indicated that the number of rewards given was significantly higher than the number of punishments. This finding aligns with Almenberg et al. [21], who discovered that when third-party observers were given the power to reward or punish, they tended to use rewards more frequently, indicating a preference for rewards. Interestingly, the number of single tokens used for punishment was significantly higher than that used for rewards, and this fluctuated drastically throughout the experiment.

This contradiction may occur for four reasons. First, people may be afraid of facing retaliation. In reality, those who punish others are often not anonymous and are at risk of retaliation themselves. In a study on cooperative behavior games, Nikiforakis and Normann [38] found that anti-social punishment and targeted revenge were common among humans. By creating models within the framework of evolutionary game theory, Rand and Nowak [39] confirmed that punishment, in some cases, was a self-interested behavior aimed at potential competitors and that anti-social punishment was widespread [21,27,39]. Therefore, people are generally more willing to use rewards than punishment for a few reasons. Second, punishment can result in losses for oneself and others, leading to reduced efficiency [26,40,41]. As a result, leaders tend to be cautious about using punishment and are reluctant to bear the costs of punishing free riders. This makes punishment less frequent. Third, when followers engage in free-riding behavior, it can evoke negative emotions and a sense of inequality in leaders [34,42], leading to increased punishment [34,42]. In contrast, rewards bring additional benefits to others and do not trigger negative emotions, making them a less risky option for leaders. Therefore, rewards are used more frequently, resulting in a smaller number of tokens being used for rewards compared to punishment. Finally, in this study, leaders were randomly assigned and may have had different cooperative preferences [29,30,31,32]. When the average contribution decreases, leaders who prefer cooperation are more likely to impose severe punishment to warn low contributors. This is consistent with studies showing that participants with a stronger cooperative inclination are more likely to implement sanctions to suppress free riding over time [37], while leaders with other preferences tend to use rewards more frequently [21,43].

The results also showed that the correlation between the leaders’ and followers’ contribution bias was negatively correlated with the number of single tokens used for rewards and positively correlated with the number of punishments. A study by Trevino (1992) [44] found that punishment was effective for most punished participants because it added risks into cost–benefit calculations. Although free riding can bring more economic benefits for free riders, the threat of punishment also puts them at a risk of loss.

According to the concept of loss aversion, people tend to be more concerned about avoiding losses than acquiring gains [45,46]. Therefore, the punishment can rapidly warn non-cooperative followers, and the reward must be more costly to make unrewarded followers aware of the relative loss of benefits.

Furthermore, some practical research has discovered that a leader’s performance is influenced by a variety of organizational variables, as well as by some personal and interpersonal characteristics [47]. Leadership effectiveness has a huge impact on the success of business initiatives, and this demonstrates the need for self-sacrificing labor that will benefit the leader’s organization [48]. Employees can be inspired by the waiver behavior of organizational leaders and use these acts to define their aspirations [47,48]. In general, good leadership actions have a major positive impact on followers and, eventually, social systems. According to certain theoretical studies, leader effectiveness has the ability to build a vision for an organization’s future, ensuring that members focus on this vision and demonstrate their commitment to the organization [49,50]. Furthermore, making personal sacrifices as a leader is one of the most straightforward ways of demonstrating a leader’s concern for the well-being of the organization [51].

The findings highlight the crucial role of leaders who use rewards and punishment to improve employee performance, as well as the need to cultivate firm engagement. These results might be seen as the extent to which the organization’s objective or task is accomplished [52]. Organizations should aim for a balance between positive and negative reinforcement. For example, a reward system could include a combination of incentives like promotions, bonuses, public recognition, etc. On the other hand, punishment could consist of a warning system or disciplinary action [53].

## 5. Conclusions

In summary, this study suggests that, while leading by example is vital, incorporating external incentives (particularly the power to punish) is crucial for maintaining high levels of cooperation in organizations.

The leader’s presence enhances the level of public cooperation by contributing more and benefiting less than the followers. However, the leader’s sacrifice of personal interests can lead to a decrease in willingness to maintain long-term cooperation. Allowing the leader to punish or reward increases and maintains public cooperation. Therefore, punishment was found to be more effective than reward in leading by example. Furthermore, we discovered that, as the difference in contributions between leaders and followers increased, the leaders tended to punish the followers more often or reward the followers with smaller amounts. This reveals that humans have different behavioral preferences for punishment and reward. This pattern of behavior suggests that humans are more motivated to avoid punishment, which explains why punishment is more effective in promoting cooperation. Additionally, the research found that this behavioral trait may be linked to the evolution of human cooperative behavior and thus requires further study.

The findings highlight some practical implications for managers in developing a new model of leadership mechanisms. First, in our study, we found that random leaders are transformational leaders, while empowered leaders are transactional. Our results show that empowered leadership significantly motivates public cooperation, supporting the notion that the best leaders are those who can lead both transformationally and transactionally [54,55,56]. Second, our findings have significant implications for further supporting and enriching leadership and strong reciprocity theory, as well as in empirically supporting research on transactional and transformational leadership, both of which have important management practical implications for promoting organizational cooperation. Finally, leading by example should be actively carried out in firms with more flat organizations, which is beneficial to perfect the authorized leadership system and enrich empowering leadership philosophy or theory.

In the future, as an extension of this research, we shall focus on different types of role modeling (leading by example) to promote cooperation in various settings, such as workplaces, communities, and online environments, as well as explore how different leadership styles coordinate with endogenous and exogenous power in order to influence group dynamics and cooperation among individuals. Nevertheless, this kind of setting includes various cultural, educational, and age backgrounds, which may limit the generalizability of the findings.

## Figures and Tables

**Figure 1 behavsci-14-00601-f001:**
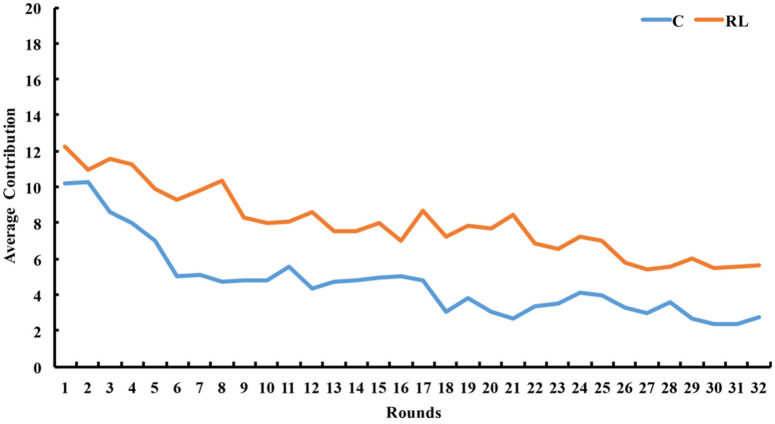
The average contribution varied between rounds in the RL and C groups. Source: authors’ compilation.

**Figure 2 behavsci-14-00601-f002:**
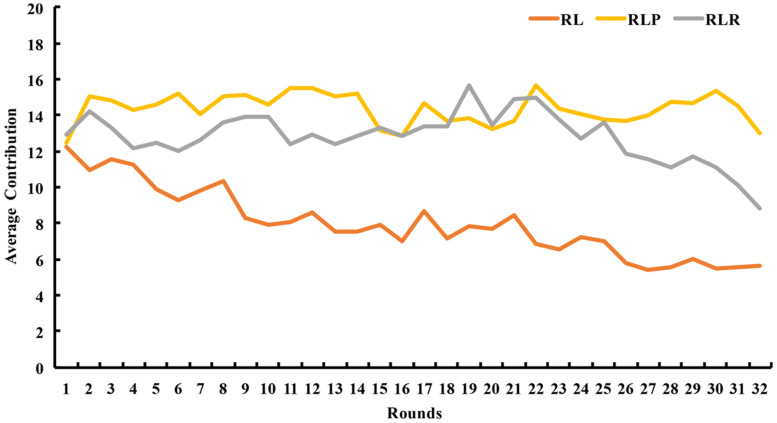
The average contribution varied between rounds in the RL group and the two groups with leadership power (RLP and RLR groups). Source: authors’ compilation.

**Figure 3 behavsci-14-00601-f003:**
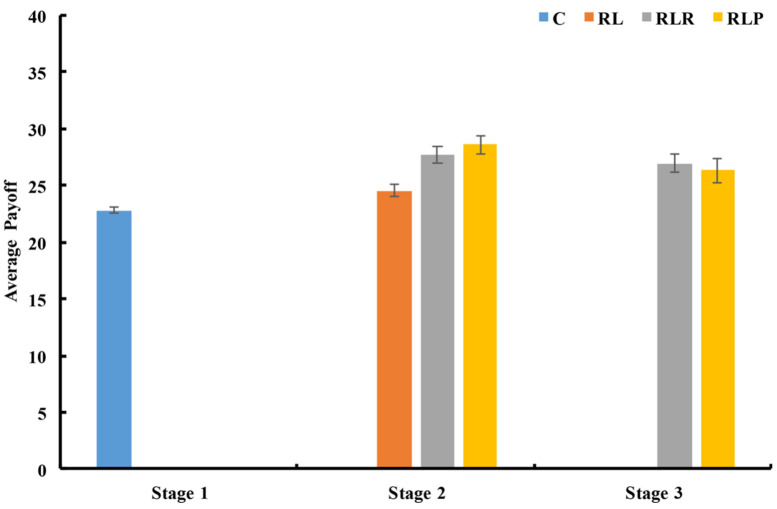
The average payoff of each group in every stage. The error bar shows one standard error. Source: authors’ compilation.

**Figure 4 behavsci-14-00601-f004:**
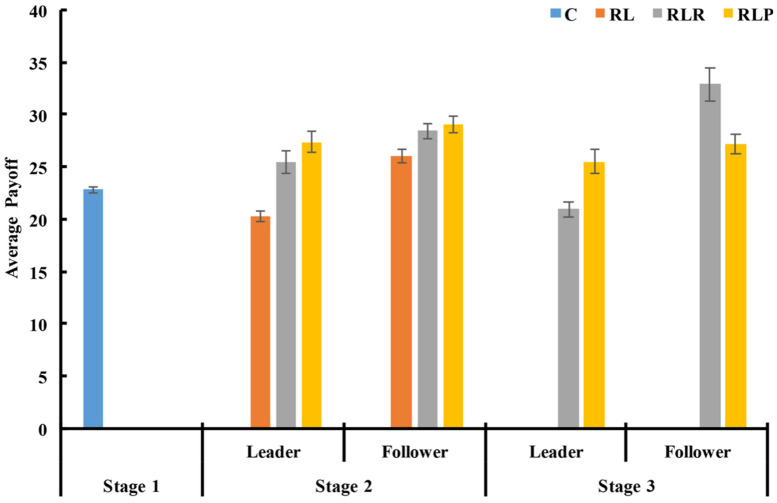
The average payoff of each role in each group in every stage. The error bar shows one standard error. Source: authors’ compilation.

**Figure 5 behavsci-14-00601-f005:**
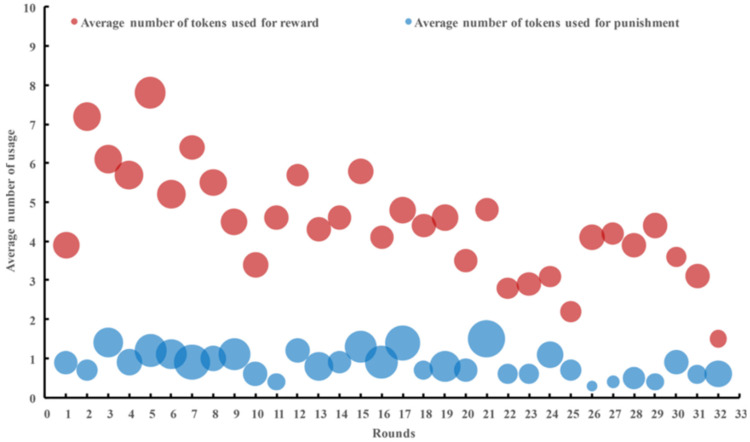
The average number of rewards and punishments used varied between rounds. The area of each bubble (reduced to 30% of the original) shows the average number of tokens used for rewards and punishments. Source: authors’ compilation.

**Figure 6 behavsci-14-00601-f006:**
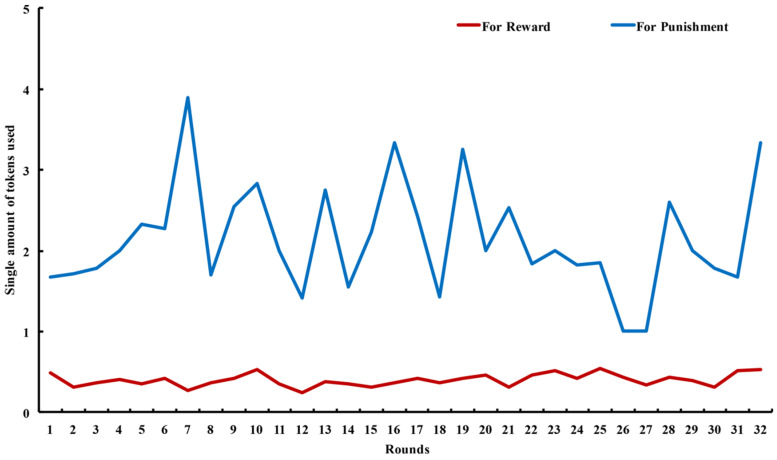
Number of single tokens used for rewards and punishments varied between rounds. Source: authors’ compilation.

**Figure 7 behavsci-14-00601-f007:**
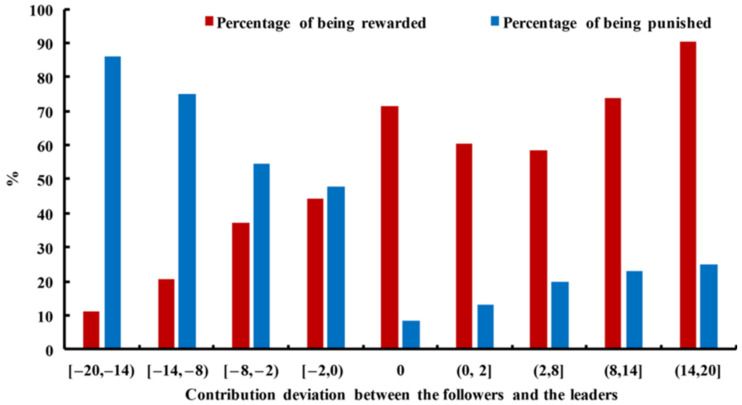
The percentage of being rewarded or punished varied with the contribution deviation between the followers and the leaders. Source: authors’ compilation.

**Table 1 behavsci-14-00601-t001:** The average contribution and payoff of the RL and the C groups (standard deviation in parenthesis).

Group	Role	Contribution	Payoff
RL	All	7.61 (2.91)	24.56 (1.74)
	Leader	11.88 (3.63)	20.30 (1.64)
	Follower	6.19 (2.75)	25.99 (1.96)
C	All	4.69 (1.58)	22.81 (0.95)

**Table 2 behavsci-14-00601-t002:** The average contribution and payoff of the three groups with leaders (standard deviation in parentheses) for all members and roles.

Group	Role	Contribution	Payoff
RLR	All	12.81 (3.89)	26.94 (2.59)
	Leader	15.08 (3.59)	20.97 (2.22)
	Follower	12.06 (4.13)	32.90 (4.97)
RLP	All	14.36 (4.20)	26.34 (3.31)
	Leader	15.60 (3.81)	25.51 (3.82)
	Follower	13.94 (4.38)	27.17 (2.88)
RL	All	7.61 (2.91)	24.56 (1.74)
	Leader	11.88 (3.63)	20.30 (1.64)
	Follower	6.19 (2.75)	25.99 (1.96)

**Table 3 behavsci-14-00601-t003:** The descriptive statistics of the number of rewards and punishments used, as well as the number of tokens used for the two powers.

	Power	Mean	Standard Deviation	Maximum	Minimum
The number of usage	Reward	65.7	37.2	146	25
	Punishment	27.1	14.65	42	3
The number of tokens used	Reward	142.7	113.59	367	26
	Punishment	59.4	41.75	147	3

## Data Availability

Data are unavailable due to privacy restrictions.

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
