# Peer review of "Can Leading by Example Alone Improve Cooperation?"

_behavsci, 2024, doi:10.3390/bs14070601_

Round 1

Reviewer 1 Report

Comments and Suggestions for Authors

The experiment is interesting and well conducted, but the results are discussed only from a theoretical point of view. On the contrary, it would be much more valuable if the results are discussed in view of their practical application inside an organization. This could be done also adding few real examples taken from the organizational literature. 

Author Response

Comments 1: The experiment is interesting and well-conducted, but the results are discussed only from a theoretical point of view. On the contrary, it would be much more valuable if the results are discussed in view of their practical application inside an organization. This could be done also adding a few real examples taken from the organizational literature.

Response 1: Thank you for pointing this out. We agreed with the reviewer’s comments. Therefore, we have rechecked the practical application inside an organization and restructured the manuscript following your comments.

Discussion

Furthermore, some practical research discovered that a leader's performance is influenced by a variety of organizational variables, as well as some personal and interpersonal characteristics[47]. Leadership effectiveness has a huge impact on the success of businesses' initiatives and demonstrates the need of self-sacrificing labor that will benefit the leader's organization[48]. Employees can be inspired by the waiver behavior of organizational leaders and use these acts to define their aspirations[47-48]. In general, good leadership actions have a major positive impact on their followers and, eventually, social systems. According to theoretical studies, leader effectiveness has the ability to build a vision for the organization's future, ensuring that members focus on this vision, and demonstrate their commitment to the organization [49-50]. Furthermore, making personal sacrifices as a leader is one of the most straightforward ways of demonstrating a leader’s concern for the well-being of the organization[51].

The findings highlight the crucial role of leaders who use rewards and punishment to improve employee performance, as well as the need to cultivate firm engagement. These results might be seen as the extent to which the organization’s objective or task is accomplished[52]. Organizations should aim for a balance between positive and negative reinforcement. For example, a reward system could include a combination of incentives like promotions, bonuses, public recognition, and more. On the other hand, punishment could consist of a warning system or disciplinary action[53].

References:

  1. Sonmez Cakir, F., & Adiguzel, Z. Analysis of leader effectiveness in organization and knowledge sharing behavior on employees and organization. Sage Open. 2020, 10, 2158244020914634.

  1. Ioan, P. Leadership and emotional intelligence: The effect on performance and attitude. Procedia Economics and Finance. 2014, 15, 985–992.

  1. Conger, J. A. (1999). Charismatic and transformational leadership in organizations: An insider’s perspective on these developing streams of research. The Leadership Quarterly. 1999, 10, 145–179.
  2. Lowe, K. B., Kroeck, K. G., & Sivasubramaniam, N. Effectiveness correlates of transformational and transactional leadership: A meta-analytic review of the MLQ literature. The Leadership Quarterly. 1996, 7, 385–425.

  1. Jacobson, C., & House, R. J. Dynamics of charismatic leadership: A process theory, simulation model, and tests. The Leadership Quarterly. 2001, 12, 75–112.

  1. Layek, D., & Koodamara, N. K. Impact of contingent rewards and punishments on employee performance: the interplay of employee engagement. F1000Research. 2024, 13, 102.

  1. Marlina, L., Setyoningrum, N. G., Mulyani, Y. S., Permana, T. E., & Sumarni, R. Improving employees working discipline with punishment, reward, and implementation of standard operational procedures. Perwira International Journal of Economics & Business. 2021, 1, 37-43.

Conclusion

In summary, the study suggests that, while leading by example is vital, incorporating external incentives, particularly the power to punish, is crucial for maintaining high levels of cooperation in organizations.

The leader's presence enhanced the level of public cooperation by contributing more and benefiting less than the followers. However, the leader's sacrifice of personal interests led to a decrease in willingness to maintain long-term cooperation. Allowing the leader to punish or reward increased and maintained public cooperation. Therefore, punishment was found to be more effective than reward in leading by example. Furthermore, we discovered that as the difference in contributions between leaders and followers increased, the leaders tended to punish the followers more often or reward the followers with smaller amounts. This reveals that humans have different behavioral preferences for punishment and reward. This pattern of behavior suggests that humans are more motivated to avoid punishment, which explains why punishment is more effective in promoting cooperation. Additionally, the research finds that this behavioral trait may be linked to the evolution of human cooperative behavior and requires further study.

Reviewer 2 Report

Comments and Suggestions for Authors

The paper describes the results of an experiment on leadership behavior in a public goods game. Results show differences in the contribution and payoff of leaders and participants depending on the setting of the group (with random leader or fixed leader, punishments or rewards).

It is difficult to assess the paper due to a lack of information on the experiment and its implementation. There is no information about the task participants had to fulfil when participating in the game. What is it about? Which information and instructions did participants get before the game? How is the role of leader defined? What is expected by a leader, what is his/her task? Furthermore, it would be interesting to know if there have been incentives for participants.

A probably interesting result is the different use of punishment and reward by leaders. However, this could also be a consequence of the definition of the role of/expectations towards leaders?

A problematic aspect of the study is that it puts not only the intervention as such in a blackbox (see lack of information above) but also the role of leaders. As a reader one gets the impression that being leader does not require any competences and qualifications. Therefore, I do not support the conclusion that the study provides insights into the complex dynamics of leadership and cooperation. The paper is lacking a critical reflection of the validity of the results of the experiment beyond the experimental setting.

Comments on the Quality of English Language

There are some typos in the text. Therefore proof reading is required.

Author Response

Comments 1: The paper describes the results of an experiment on leadership behavior in a public goods game. Results show differences in the contribution and payoff of leaders and participants depending on the setting of the group (with random leader or fixed leader, punishments or rewards). It is difficult to assess the paper due to a lack of information on the experiment and its implementation. There is no information about the task participants had to fulfill when participating in the game. What is it about? Which information and instructions did participants get before the game? How is the role of a leader defined? What is expected by a leader, what is his/her task? Furthermore, it would be interesting to know if there have been incentives for participants. A probably interesting result is the different use of punishment and reward by leaders. However, this could also be a consequence of the definition of the role of/expectations towards leaders. A problematic aspect of the study is that it puts not only the intervention as such in a black box (see lack of information above) but also the role of leaders. As a reader, one gets the impression that being leader does not require any competences and qualifications. Therefore, I do not support the conclusion that the study provides insights into the complex dynamics of leadership and cooperation. The paper is lacking a critical reflection of the validity of the results of the experiment beyond the experimental setting.

Response 1: Thank you very much for your careful comments. We have rewritten the manuscript following the reviewer’s comments.

Appendices A, B, and C provide information on the experiment and its implementation interfaces for the control (C) group, the random leader (RL) group, and the random leader punishment (RLP)/ random leader reward (RLR) groups, respectively.

Appendix A. The control (C) group instructions and experimental interface.

This is a decision-making experiment, and if you follow the instructions carefully, you can make money from your decisions. Please read these instructions carefully.

  1. Your instructions are only for personal reading, and you were not allowed to communicate with the other participants during the experiment. Please get in touch with us if you have any questions. If you violate this policy, you will be removed from the experiment without compensation.

  1. In the experiment, you will be given a certain number of tokens rather than cash, and all of your gains will be calculated with the tokens. At the end of the experiment, the tokens obtained can be exchanged for actual cash, with 1 chip equaling 0.015 RMB. 

  2. The decision determined how many tokens could be obtained during the experiment. In the experiment, you and three other participants will make up a group to play an investment game together, and each member of the group will be assigned a number (1 to 4) to represent his or her identity. The group members and their numbers were fixed for the duration of the experiment.

  1. The experiment consisted of 32 rounds; in each round, each group member was given 20 initial tokens. You need to decide how many tokens to put in the public project, and the tokens not put in the public project will be retained by you. The total amount of tokens that the four members of the group put into the public project will be multiplied by 1.6 and divided equally among each member of the group. Therefore, the tokens you gain in each round are 20 - (tokens you put into the public project) +1.6 x (total in the public project) /4.

  1. Here, we present two examples to understand the payoff-calculation method better.

Example 1: If each group member gives 20 tokens to a public project, then each group member receives 32 tokens: 20-20+1.6× (20+20+20+20) /4.

Example 2: If three group members give 20 tokens in the public project, and the other group member does not give a token in the public project, then the group members who give 20 tokens will receive 24 tokens: 20-20+1.6× (20+20+20+0) /4, and the group member who does not give tokens will receive 44 tokens: 20-0+1.6 x (20+20+20+0) /4.

  1. At the end of each round, you will get feedback on the number of tokens you give in that round, the total number of tokens all group members give in that round, and your payoff for that round. After that, you will move on to the next round.

  1. See the experimental interface diagram for details. Your total payoff in the experiment is the sum of the tokens you earned in each of the 32 rounds that can be exchanged for cash at the end of the experiment.

Once the experiment begins, as shown in the image below, you will have 30 seconds to make a decision, enter the number of tokens you want to put into the public project (between 0 and 20) in the box and click "OK".

  1. When other group members have not clicked "OK", you will enter the waiting screen

(as shown below).

  1. When the other members of the group click "OK" to enter the next interface(as shown below), then you will see the number of tokens you give in this round, the total amount of tokens given by all the members of the round, and your payoff for this round. The display time of this interface is 30 seconds, you can wait or click "continue" to enter the next round.

Appendix B. Random leader  (RL) group instructions and experimental interface.

This is a decision-making experiment, and if you follow the instructions carefully, you can make money from your decisions. Please read these instructions carefully.
1. Your instructions are only for personal reading, and you were not allowed to communicate with the other participants during the experiment. Please get in touch with us if you have any questions. If you violate this policy, you will be removed from the experiment without compensation.

  1. In the experiment, you will be given a certain number of tokens rather than cash, and all of your gains will be calculated with the tokens. At the end of the experiment, the tokens obtained can be exchanged for actual cash, with 1 chip equaling 0.015 RMB. 

  1. The decision determined how many tokens could be obtained during the experiment. In the experiment, you and three other participants will make up a group to play an investment game together, and each member of the group will be assigned a number (1 to 4) to represent his or her identity. The group members and their numbers were fixed for the duration of the experiment.

  1. The experiment consisted of 32 rounds; in each round, each group member was given 20 initial tokens. You need to decide how many tokens to put in the public project, and the tokens not put in the public project will be retained by you. The total amount of tokens that the four members of the group put into the public project will be multiplied by 1.6 and divided equally among each member of the group. Therefore, the tokens you gain in each round are 20 - (tokens you put into the public project) +1.6 x (total in the public project) /4.

  1. Here, we present two examples to understand the payoff-calculation method better:

Example 1: If each group member gives 20 tokens to a public project, then each group member receives 32 tokens: 20-20+1.6× (20+20+20+20) /4.

Example 2: If three group members give 20 tokens in the public project, and the other group member does not give a token in the public project, then the group members who give 20 tokens will receive 24 tokens: 20-20+1.6× (20+20+20+0) /4, and the group member who does not give tokens will receive 44 tokens: 20-0+1.6 x (20+20+20+0) /4.

  1. Each round of this experiment consisted of two stages.

Stage 1: The system randomly selects a group member to be the first to give tokens in public projects(defined as the leader). The first stage of each round was re-randomly selected.

Stage 2: The other three members of the group, after knowing the amount of tokens given by the leader put tokens into the public project.

  1. At the end of each round, you will get the following feedback: the number of tokens you give in that round, the total amount of tokens all group members give in that round, and your payoff for this round. After that, you will move on to the next round.

  1. See the experimental interface diagram for details. Your total payoff in the experiment is the sum of the tokens you earned in each of the 32 rounds that can be exchanged for cash at the end of the experiment.

  1. When the experiment begins, as depicted in the image below, you will receive your number. Click "OK" to proceed to the next interface.

  1. When other group members have not clicked 'OK', you will enter the waiting interface, as shown below the image.

  1. When other group members click "OK," they will be taken to the next screen (see below). The page in this interface will show the number of members who were the first to contribute tokens to the public project (definedas the leader). Click "OK" to move on to the next interface.

  1. When all group members click "OK," the leader will see the following interface, while the other group members will enter the waiting screen, waiting for the leader to add tokens to the public project. The leader will have 30 seconds to make a decision before clicking "OK" to proceed to the next interface.

  1. At this point, the screen displays the number of tokens that the leader has givenin the public project, and the remaining three members of the group have 30 seconds to make a decision and click "OK".

  1. When all three members of the group click "OK", you will be taken to the interface below, where you will see the number of tokens you gave in this round, the total number of tokens given by all members in this round, and your payoff for that round. This screen's display time is 30 seconds. To proceed to the next interface, you can either wait or click "Continue".

Appendix C. The random leader punishment (RLP) /the random leader reward (RLR) group instructions and experimental interface.

This is a decision-making experiment, and if you follow the instructions carefully, you can make money from your decisions. Please read these instructions carefully.

  1. Your instructions are only for personal reading, and you were not allowed to communicate with the other participants during the experiment. Please contact us if you have any questions. If you violate this policy, you will be removed from the experiment without compensation.

  1. In the experiment, you will be given a certain number of tokens rather than cash, and all of your gains will be calculated with the tokens. At the end of the experiment, the tokens obtained can be exchanged for actual cash, with 1 chip equaling 0.015 RMB. 

  2. The decision determined how many tokens could be obtained during the experiment. In the experiment, you and three other participants will make up a group to play an investment game together, and each member of the group will be assigned a number (1 to 4) to represent his or her identity. The group members and their numbers were fixed for the duration of the experiment.

  1. The experiment consisted of 32 rounds; in each round, each group member was given 20 initial tokens. You need to decide how many tokens to put in the public project, and the tokens not put in the public project will be retained by you. The total amount of tokens that the four members of the group put into the public project will be multiplied by 1.6 and divided equally among each member of the group. Therefore, the tokens you gain in each round are 20 - (tokens you put into the public project) +1.6 x (total in the public project) /4.
  2. Here, we present two examples to understand the payoff-calculation method better:

Example 1: If each group member gives 20 tokens to a public project, then each group member receives 32 tokens: 20-20+1.6× (20+20+20+20) /4.

Example 2: If three group members give 20 tokens in the public project, and the other group member does not give a token in the public project, then the group members who give 20 tokens will receive 24 tokens: 20-20+1.6× (20+20+20+0) /4, and the group member who does not give tokens will receive 44 tokens: 20-0+1.6 x (20+20+20+0) /4.

  1. Each round of this experiment consisted of three stages:

Stage 1: The system randomly selects a group member to be the first to give tokens in public projects(defined as the leader). The first stage of each round was re-randomly selected.

Stage 2: The other three members of the group, after knowing the amount of tokens given by the leader, put tokens into the public project.

Stage 3: The leader sees the tokens given by the other members to decide whether to spend the tokens gained from the first two stages to reduce/add the tokens of the other members. Every token that he/she uses can reduce/add the corresponding member by three tokens.

  1. See the experimental interface diagram for details.

 Your total payoff in the experiment is the sum of the tokens you earned in each of the 32 rounds that can be exchanged for cash at the end of the experiment. When the experiment begins, as depicted in the image below, you will receive your number. Click "OK" to proceed to the next interface.

  1. When other group members have not clicked 'OK', you will enter the waiting interface, as shown below the image.

  1. When other group members click "OK," they will be taken to the next screen (see below). The page in this interface will show the number of members who were the first to contribute tokens to the public project (defined as the leader). Click "OK" to move on to the next interface.

  1. When all group members click "OK," the leader will see the following interface, while the other group members will enter the waiting screen, waiting for the leader to add tokens to the public project. The leader will have 30 seconds to make a decision before clicking "OK" to proceed to the next interface.

  1. At this point, the screen displays the number of tokens that the leader has given in the public project, and the remaining three members of the group have 30 seconds to make a decision and click "OK".

  1. When all three members of the group click "OK", you will be taken to the interface below, where you will see the number of tokens you gave in this round, the total number of tokens given by all members in this round, and your payoff for that round. This screen's display time is 30 seconds. To proceed to the next interface, you can either wait or click "Continue".

  1. The leader will enter the following interface after all other group members select "Continue" (other group members will enter the waiting interface). The leader will review the tokens of other group members and determine whether to use the tokens earned in the first two phases to reduce or add the tokens of other group members. Every token utilized by the leader can reduce or add three tokens to the corresponding group members.

  1. Clicking "OK" will display the interface below for the group leader and other participants. It will show the quantity of tokens utilized in the third stage, the amount of tokens increased or decreased, and the payoff at the end of the round. Clicking "Continue" allows the entire group to move on to the next round.

Discussion

Furthermore, some practical research discovered that a leader's performance is influenced by a variety of organizational variables, as well as some personal and interpersonal characteristics[47]. Leadership effectiveness has a huge impact on the success of businesses' initiatives and demonstrates the need of self-sacrificing labor that will benefit the leader's organization[48]. Employees can be inspired by the waiver behavior of organizational leaders and use these acts to define their aspirations[47-48]. In general, good leadership actions have a major positive impact on their followers and, eventually, social systems. According to theoretical studies, leader effectiveness has the ability to build a vision for the organization's future, ensuring that members focus on this vision, and demonstrate their commitment to the organization [49-50]. Furthermore, making personal sacrifices as a leader is one of the most straightforward ways of demonstrating a leader’s concern for the well-being of the organization[51].

The findings highlight the crucial role of leaders who use rewards and punishment to improve employee performance, as well as the need to cultivate firm engagement. These results might be seen as the extent to which the organization’s objective or task is accomplished[52]. Organizations should aim for a balance between positive and negative reinforcement. For example, a reward system could include a combination of incentives like promotions, bonuses, public recognition, and more. On the other hand, punishment could consist of a warning system or disciplinary action[53].

References:

  1. Sonmez Cakir, F., & Adiguzel, Z. Analysis of leader effectiveness in organization and knowledge sharing behavior on employees and organization. Sage Open. 2020, 10, 2158244020914634.

  1. Ioan, P. Leadership and emotional intelligence: The effect on performance and attitude. Procedia Economics and Finance. 2014, 15, 985–992.

  1. Conger, J. A. (1999). Charismatic and transformational leadership in organizations: An insider’s perspective on these developing streams of research. The Leadership Quarterly. 1999, 10, 145–179.
  2. Lowe, K. B., Kroeck, K. G., & Sivasubramaniam, N. Effectiveness correlates of transformational and transactional leadership: A meta-analytic review of the MLQ literature. The Leadership Quarterly. 1996, 7, 385–425.

  1. Jacobson, C., & House, R. J. Dynamics of charismatic leadership: A process theory, simulation model, and tests. The Leadership Quarterly. 2001, 12, 75–112.

  1. Layek, D., & Koodamara, N. K. Impact of contingent rewards and punishments on employee performance: the interplay of employee engagement. F1000Research. 2024, 13, 102.

  1. Marlina, L., Setyoningrum, N. G., Mulyani, Y. S., Permana, T. E., & Sumarni, R. Improving employees working discipline with punishment, reward, and implementation of standard operational procedures. Perwira International Journal of Economics & Business. 2021, 1, 37-43.

Conclusion

In summary, the study suggests that, while leading by example is vital, incorporating external incentives, particularly the power to punish, is crucial for maintaining high levels of cooperation in organizations.

The leader's presence enhanced the level of public cooperation by contributing more and benefiting less than the followers. However, the leader's sacrifice of personal interests led to a decrease in willingness to maintain long-term cooperation. Allowing the leader to punish or reward increased and maintained public cooperation. Therefore, punishment was found to be more effective than reward in leading by example. Furthermore, we discovered that as the difference in contributions between leaders and followers increased, the leaders tended to punish the followers more often or reward the followers with smaller amounts. This reveals that humans have different behavioral preferences for punishment and reward. This pattern of behavior suggests that humans are more motivated to avoid punishment, which explains why punishment is more effective in promoting cooperation. Additionally, the research finds that this behavioral trait may be linked to the evolution of human cooperative behavior and requires further study.

Round 2

Reviewer 2 Report

Comments and Suggestions for Authors

The authors have analysed the comments and tried to take them into account when revising the paper. Many thanks for this. The role of leadership is now addressed in the concluding section, but in my opinion this only partially solves the underlying problem. In my opinion, the results are not sufficiently reflected to draw conclusions beyond the present experiment. The article can be seen as a contribution to the methodological discussion about experiments, but not to the discussion about leadership.

Comments on the Quality of English Language

The authors have analysed the comments and tried to take them into account when revising the paper. Many thanks for this. The role of leadership is now addressed in the concluding section, but in my opinion this only partially solves the underlying problem. In my opinion, the results are not sufficiently reflected to draw conclusions beyond the present experiment. The article can be seen as a contribution to the methodological discussion about experiments, but not to the discussion about leadership.

Author Response

Comments 1: The authors have analyzed the comments and tried to take them into account when revising the paper. Many thanks for this. The role of leadership is now addressed in the concluding section, but in my opinion this only partially solves the underlying problem. In my opinion, the results are not sufficiently reflected to draw conclusions beyond the present experiment. The article can be seen as a contribution to the methodological discussion about experiments, but not to the discussion about leadership.

Response 1: Thank you for your comments. We have revised the manuscript and addressed it accordingly.

The findings highlight some practical implications for managers in developing a new model of leadership mechanisms. First, in our study, we found that random leaders are transformational leaders, while empowered leaders are transactional. Our results show that empowered leadership significantly motivates public cooperation, supporting that the best leaders are those who can lead both transformationally and transactionally[54-56]. Second, Our findings have significant implications for further supporting and enriching leadership and strong reciprocity theory, as well as empirically supporting research on transactional and transformational leadership, both of which have important management practical implications for promoting organizational cooperation. Finally, Leading by example should be actively carried out in firms with more flat organizations, which is beneficial to perfect the authorized leadership system and enrich the empowering leadership philosophy or theory.

References:

  1. Bass, B. M. Two decades of research and development in transformational leadership. European Journal of Work and Organizational Psychology.1999, 8, 9-32.

  1. Judge, T. A., & Piccolo, R. F. Transformational and Transactional Leadership: A Meta-Analytic Test of Their Relative Validity. Journal of Applied Psychology.2004, 89, 755-768.

  1. Cho, Y., Shin, M., Billing, T. K., & Bhagat, R. S. Transformational leadership, transactional leadership, and affective organizational commitment: a closer look at their relationships in two distinct national contexts. Asian Business & Management. 2019, 18, 187-210.